# Label-Free SERS and MD Analysis of Biomarkers for Rapid Point-of-Care Sensors Detecting Head and Neck Cancer and Infections

**DOI:** 10.3390/s23218915

**Published:** 2023-11-02

**Authors:** Edoardo Farnesi, Silvia Rinaldi, Chen Liu, Jonas Ballmaier, Orlando Guntinas-Lichius, Michael Schmitt, Dana Cialla-May, Juergen Popp

**Affiliations:** 1Institute of Physical Chemistry (IPC) and Abbe Center of Photonics (ACP), Friedrich Schiller University Jena, Member of Leibniz Centre for Photonics in Infection Research (LPI), Helmholtzweg 4, 07743 Jena, Germany; edoardo.farnesi@uni-jena.de (E.F.); chen.liu@leibniz-ipht.de (C.L.); m.schmitt@uni-jena.de (M.S.); juergen.popp@uni-jena.de (J.P.); 2Leibniz Institute of Photonic Technology, Member of Leibniz Health Technologies, Member of Leibniz Centre for Photonics in Infection Research (LPI), Albert-Einstein-Straße 9, 07745 Jena, Germany; 3Institute for the Chemistry of Organo Metallic Compounds, National Research Council of Italy (CNR), Via Madonna del Piano 10, Sesto Fiorentino, 50019 Florence, Italy; silvia.rinaldi@iccom.cnr.it; 4Department of Otorhinolaryngology-Head and Neck Surgery, Jena University Hospital, 07747 Jena, Germany; jonas.ballmaier@med.uni-jena.de (J.B.); orlando.guntinas@med.uni-jena.de (O.G.-L.)

**Keywords:** biomarkers, biosensors, cancer, infections, molecular dynamics, point-of-care, saliva, SERS

## Abstract

For the progress of point-of-care medicine, where individual health status can be easily and quickly monitored using a handheld sensor, saliva serves as one of the best-suited body fluids thanks to its availability and abundance of physiological indicators. Salivary biomarkers, combined with rapid and highly sensitive detection tools, may pave the way to new real-time health monitoring and personalized preventative therapy branches using saliva as a target matrix. Saliva is increasing in importance in liquid biopsy, a non-invasive approach that helps physicians diagnose and characterize specific diseases in patients. Here, we propose a proof-of-concept study combining the unique specificity in biomolecular recognition provided by surface-enhanced Raman spectroscopy (SERS) in combination with molecular dynamics (MD) simulations, which give leave to explore the biomolecular absorption mechanism on nanoparticle surfaces, in order to verify the traceability of two validated salivary indicators, i.e., interleukin-8 (IL-8) and lysozyme (LYZ), implicated in oropharyngeal squamous cell carcinoma (OSCC) and oral infection. This strategy simultaneously assures the detection and interpretation of protein biomarkers in saliva, ultimately opening a new route for the evolution of fast and accurate point-of-care SERS-based sensors of interest in precision medicine diagnostics.

## 1. Introduction

Early detection and precision medicine are fundamental paradigms that improve patient prognosis and enhance the success of rapid therapy. These aspects are crucial in the successful treatment of important and widespread diseases such as cancer, bacterial infections, cardiovascular diseases, and diabetes mellitus, thus preventing complications and boosting the probability of survival and the life quality of the patient [1].

In this scenario, point-of-care diagnostics represents the research branch where cutting-edge technologies are tested with the aim of providing rapid, sensitive, and suitable medical information, avoiding the time delays associated with traditional protocols for collecting and transporting patient samples into centralized labs performing gold-standard methods such as high-performance liquid chromatography (HPLC)-based approaches combined with mass spectrometry [2,3,4]. The ultimate goal is the development of portable and easy-to-use biosensors for health monitoring, early detection, and specific treatments based on the patient’s characteristics [5,6]. However, the challenge in point-of-care diagnostics for personalized medicine is determining which biomarkers to test and estimating their concentration using a suitable technique. Consequently, access to high-quality measurable nano-indicators of a healthy or irregular physiological status, which allow for estimating any alteration in human health conditions, is required [7,8].

For example, portable glucose meters for monitoring blood sugar levels in diabetes patients and early pregnancy test kits are important achievements that have been a driving force in the development of personalized point-of-care medicine [9,10]. An important goal to pave the way for simple access to point-of-care diagnostics is related to the development of liquid biopsy diagnostic devices with the attempt to provide valid, non-invasive, and reliable assays identifying biomarkers from body fluids. 

Among the possible non-invasively collectible biofluids, saliva is one of the best candidates, as it is easily accessible and easy to transport and store. Further, saliva is considered the “mirror of the body” due to its continuous renovation and its proven richness in biomarkers that are dysregulated in several diseases [11].

Recent studies investigating the use of surface-enhanced Raman scattering (SERS) using plasmonic metal nanoparticles interacting with biomarkers for biofluid characterization to enhance the molecular specific Raman fingerprint information by 4 to 8 orders of magnitude have shown promising results [12,13]. The enhancement in the Raman signal overpasses the sensitivity issues that have limited the application of Raman spectroscopy in disease diagnosis so far with the attempt to identify cancer-related biomarker concentration changes in urine, serum, plasma and saliva. Differences in the occurrence of Raman peaks and peak intensity ratios have been found between patients and healthy people, allowing for the non-invasive detection of lung cancer, cervical cancer and other types of diseases [14,15,16,17,18,19,20,21,22]. Moreover, by applying a portable Raman spectrometer for read-out, the SERS technique is available at the point-of-care [23,24,25]. 

However, attempts to characterize and discriminate proteins in biological fluids using the SERS approach is still a challenging task. Indeed, difficulties in spectral interpretation caused by an unpredictable enhancement in components and bands overlapping are still hampering its wide application. Combining SERS biosensing techniques with molecular dynamics (MD) can be an effective strategy to deal with the complexity of the SERS dataset. Notably, MD has proven to be successful in correlating SERS spectra with protein structural signatures and detecting the conformational modulation of protein interfaces [26,27,28]. All-atom MD computations are in fact powerful tools for the atomistic characterization of proteins and offer a thorough understanding of the molecular drivers behind protein adsorption on the metal surfaces’ highlighting structural hotspot that accounts for SERS signal differences [29]. In this scenario, the combined SERS-MD approach will be used to disentangle specific disease-related biomarkers present in a biological fluid.

Within this paper, we aim to report on salivary biomarkers analyzed with such a combined SERS-MD approach to characterize two important proteins found in saliva, which were not, to the best of our knowledge, described before. Currently detected predominantly using an expensive enzymatic immunoassay (i.e., ELISA test) [30], lysozyme (LYZ) and interleukin-8 (IL-8) have already been shown to be differentially expressed in several disease states of head and neck cancer or infections. IL-8 (1–3 ng/mL) is a non-structural protein secreted by neutrophils and macrophages, which plays an important role in the proliferation of angiogenesis and, in particular, its concentration dramatically increases in the presence of oropharyngeal squamous cell carcinoma (OSCC), which is one type of head and neck cancer [31,32,33,34]. On the other hand, LYZ (0.5–4 ng/mL) is a muramidase protein that is part of the innate salivary defense mechanisms, resulting in high concentration levels in cases of low immunity conditions [35,36,37]. It is present in saliva mainly originating from the major and minor salivary glands and it exerts enzymatic activity via hydrolysis of the bacterial wall causing cell lysis. Overall, this pilot study, by combining SERS and MD techniques, showcases the needed tool to address the challenging recognition and interpretation of disease-specific biomarkers in body fluids, paving the way to novel, fast and accurate point-of-care SERS-based devices for healthcare monitoring.

## 2. Methods

Saliva collection and sample preparation. Whole saliva samples were collected from three healthy individuals. This study was approved by the Jena University Hospital Ethics Committee (RAMTUMAR study; no. 2021-2305-Material), and all volunteers provided written informed consent for enrolling in this study (Appendix A). About 2 mL of saliva, after a congruous lag time from feeding and teeth brushing, were collected for each case in the morning. Only samples with no sign of blood contamination after visible inspection were used for measurement. Before saliva collection, the healthy volunteers were asked to rinse their mouths with water three times. The samples were kept at −80° C until the day of the SERS analysis. Then, after thawing at room temperature, the saliva was transferred to a 1.5 mL centrifuge tube, and it was centrifuged for 20 min at 9000 g to remove the oral mucous epithelial cells and other debris, obtaining pure saliva. An aliquot of 0.5 mL was filtered with a 30 k cut-off (Amicon Ultracel, Sigma-Aldrich, St. Louis, MO). Silver-coated silicon nanopillar substrates (SERStrates™), purchased from Silmeco ApS (Copenhagen, Denmark) [38,39], were incubated with both unfiltered and filtered supernatants for 20–25 min in 1 mL tubes. After incubation, the SERS chips were placed on a microscope slide and left to dry slowly at room temperature. This procedure allows for the leaning of the silver nanopillars for a most intense SERS response and minimizes the molecular rearrangements during the absorption of biomarkers on the silver surface in order to approximate the simulated solvent conditions as closely as possible within the limitations of MD.

Instrumentation and spectral analysis. SERS measurements were performed using a WITec alpha300 R confocal Raman microscope (Ulm, Germany) equipped with a 785 nm laser. The Raman scattered light was collected with a 10× objective lens. The laser light incident on the sample was 5 mW. SERS spectra were recorded using a 600 groove/mm grating and an integration time of 5 s with one accumulation. The acquisition was performed over a 1.5 × 1.5 mm^2^ area in the center of the chip with a 250-point map. Spectral analysis was performed with Origin Pro 2022 (OriginLab, Northampton, MA, USA) software. After removing outliers, the raw spectra, acquired in the range of 600 cm^−1^ to 2200 cm^−1^, were processed with baseline correction (fifth-order polynomial fit), smoothing, and normalization by area (see Appendix A for the background-corrected SERS spectra). Mean intensity profiles and standard deviation were obtained for each sample. All average spectra were normalized again to the Raman peak at 2126 cm^−1^, typically assigned to C-N stretching vibrations of the thiocyanate ion present in saliva. Band assignment was conducted based on data from the literature [16,40,41,42,43]. 

MD setup and analyses. MD simulations were performed using the Gromacs v2020 package [44] by applying the CHARMM27 force field (FF) [45,46], whereas Ag surfaces were simulated using the AgP-CHARMM FF developed by Hughes et al. [47], which recovers the electric metal polarization that classical FFs do not take into account. Silver surfaces were modeled with a five slab of Ag atoms with symmetry according to the experimental validation of SERS substrates and previous computational studies [27,48,49]. LYS and IL-8 crystallographic structures were retrieved from the RCSB databank [50] (PDB ID: 2LZT and 3IL8, respectively). Two replicas of MD simulations (>200 ns), both in the presence and in the absence of the Ag surfaces, were performed for each protein. To maximize the conformational variability explored upon the adsorption on the SERS substrate, three different random orientations of the proteins with respect to the silver slab were used in each simulation. Proteins were located over the silver surface considering a minimum distance of >1.0 nm to avoid any possible initial bias. The systems were solvated in a simulation box of explicit water molecules (TIPS3P model) [51], counterions were added to neutralize the system, and periodic boundary conditions were imposed treating the silver slab as a period molecule to simulate an infinite surface. After minimization using the steepest descent algorithm, the systems were subjected to an equilibration phase with protein-heavy atoms restrained, and then unrestrained for 1 ns each, in an NVT ensemble with the temperature controlled using a velocity-rescaling thermostat. Then, a 1 ns NPT equilibration run was performed by controlling the pressure with a Berendsen barostat (1 atm). Finally, NPT production runs with the time step set to 2.0 fs were performed using a Parrinello–Rhaman barostat. Electrostatic energies were evaluated using the particle mesh Ewald method [52] and Lennard–Jones forces with a cut-off of 1 nm. All bonds involving hydrogen atoms were constrained using the LINCS algorithm [53]. Analysis of the simulations was performed using a combination of standard gromacs utilities. Cluster analysis was carried out using the gmx cluster module of gromacs to evaluate the most representative binding poses on the silver surface of the two protein models. Clustering was obtained with the gromos method fitting heavy atoms of the protein–silver complexes (RMSD cutoff of 0.45 nm). The protein–surface interaction energies (E_int_) were determined using the Lennard–Jones potential component of the non-bonded energies computed between the silver atoms and protein residues. E_int_ calculated using the gromacs rerun option was computed for the whole protein on the most representative structure resulting from cluster analysis, while average values per residue were obtained across the simulation trajectories. 

## 3. Results

SERS analysis. Using a well-established, rapid, and routinely achievable sample preparation procedure (see Section 2 above) pure saliva specimens, avoiding food debris and blood contaminations were obtained.

In the first part of this study, we evaluated the SERS signals of unfiltered and filtered saliva incubated with silver SERS nanopillar substrates. The NIR wavelength of 785 nm was found to be the optimal Raman excitation wavelength for the SERS-based detection of the biomarker mixture in saliva, according to a previous experiment reported in the literature [21]. To reach our purpose of identifying Raman bands attributable to the salivary biomarkers IL-8 and LYZ, having a molecular weight of 8 and 15 kDa, respectively, the samples were filtered with a membrane cut-off of 30 kDa, guaranteeing the ability to acquire insight on such small proteins. Also, this approach overcomes the SERS signal loss due to the adsorption of weighty proteins (i.e., human serum albumin), protein aggregates, or oral cell debris contained in saliva, which may hinder the formation of hotspots on the SERS chips.

In Figure 1, average SERS spectra of filtered and unfiltered saliva samples for the fingerprint Raman wavenumber region (600–1800 cm^−1^) are compared. The collected SERS spectra show a good signal-to-noise ratio (see also Appendix A) with no background contribution from the applied SERS substrates. Following the literature in the field, the most common SERS bands can be assigned to amino acids and proteins [40,41,54].

The filtering process with a cut-off of 30 kDa does not lead to a loss of information, resulting only in changes in peak ratios within the wavenumber areas around 1650 cm^−1^, assigned to amide I, 1250 cm^−1^, due to amide III, 1000 cm^−1^, related to ring breathing vibrations and 720 cm^−1^, assigned to COO^−^ deformation vibrations. These results indicate that the weighty protein retention ensures an easier adsorption of small proteins onto the silver SERS nanopillar surfaces [14]. In Appendix A, 250 SERS spectra of each saliva sample, plotted in the wavenumber range of 600–2200 cm^−1^, illustrate that the most prominent peak results at 2126 cm^−1^. Based on the molecular vibrational assignment found in the literature, this band may result as an indicator of the presence of thiocyanate in saliva [16,55,56]. The latter may enter the human body through the diet, particularly with some vegetables, cheese and milk, but can also be traceable in smokers’ saliva; in fact, the thiocyanate ion can be used as an indicator of tobacco smoke exposure.

MD. Atomistic molecular dynamics (MD) calculations can provide atomic-level details of a protein’s structure over time, showing structural changes resulting from the interaction with the nanomaterial. Thus, extensive MD simulations were performed in the presence of a silver slab using three different starting protein-PP orientations (see Section 2). Proteins were also simulated in the absence of the silver layer using the same setup to highlight possible structural rearrangements upon protein binding. For both the LYZ and IL-8 systems, the presence of the silver nanosurface induces small conformational rearrangements in the simulated timescale, as highlighted by the slight increase in the average root mean square deviation of atomic positions (see Appendix A). This is also confirmed by the analysis of the radius of gyration (see Appendix A), which is a rough measure of the compactness of a structure, providing a proxy for the spreading of the proteins on the silver layers. Finally, cluster analyses (see Section 2) were performed on the protein-Ag interface to obtain the most representative binding conformations populated along the MD simulations shown in Figure 2. In agreement with the parameters described above, a substantial invariance in the 3D structure of the proteins in both cases was observed, a behavior which is also consistent with the results of previous studies [57,58] on analog systems and reinforces the reliability of our model. Taken together, these findings suggest that biosensors are capable of recording the native conformations of proteins, without altering biomarker structures. Similarly, the secondary structure of both proteins is not significantly affected by the presence of silver (see Table 1). However, the alpha and beta folds behave differently, with the former exhibiting the largest deformation. Consequently, lysozyme, which possesses a higher content of alpha-fold structure, displays greater sensitivity to interactions as compared with IL-8.

Next, we examined the energetics that drive the adsorption process (see Section 2). The interaction energies calculated on the representative structures obtained from the cluster analysis (see Section 2 and Table 1) pinpoint a similar binding mode for the two systems, which is directly related to the number of contacts (i.e., amino acid engaged into the interaction) the two proteins can make with the silver surface. 

Finally, to highlight possible drivers of adsorption, the contribution of each type of amino acid to the stabilization of the interaction with the surface was analyzed, both energetically and dynamically (see Figure 3). In both systems, polar and charged amino acids are the main interactors. In particular, amino acids with large side chains (e.g., ARG and LYS) stabilize the complex by increasing the contact area and thereby aligning parallel to the Ag surface. There are, however, differences between the two systems. In the case of LYZ, the main drivers are the charged arginines on the protein surface, which take into account 32% and 40% of the interaction energy, respectively and the dynamical contacts with the silver surface. On the other hand, in the IL-8 system, an important role is played by asparagines in addition to charged amino acids (e.g., GLU, ARG, and LYS).

Correlation between SERS and MD data. Data obtained from 30 kDa-filtered saliva SERS spectra and MD simulations of IL-8 and LYZ absorbed on the silver slab were correlated to select which SERS peaks can be attributed to the vibrational modes of the amino acids engaged in the stabilization of the interaction with the silver surface. For our purpose, the most important peaks in the mean SERS spectrum of the saliva samples (illustrated in Figure 4), showing reproducible and well-resolved bands, can be found at 728 cm^−1^ (deformation of COO^−^), 1000 cm^−1^ (ring breathing vibrations), 1325 cm^−1^ (deformation of CH_2_), 1444 cm^−1^ (deformation NH), and 1650 cm^−1^ (amide-I) (see also Table 2 for a detailed band assignment). In Table 2, the band assignment around 1000 cm^−1^ is not considered because this wavenumber region is mainly dominated by vibrational modes due to aromatic amino acids, e.g., phenylalanine and tryptophan [54]. From our MD calculation, we found that aromatic amino acids have no indicative role during the interaction between the analyzed proteins and the Ag surface. 

The MD calculation aimed to unravel the main amino acids involved in the adsorption of IL-8 and LYZ on the silver surface: arginine (ARG), glutamic acid (GLU), asparagine (ASN), glutamine (GLN), and lysine (LYS) majorly stabilize the interaction (see Table 1 and Figure 3). Regarding LYZ, ARG results as the main driver within the adsorption process, and its specific SERS vibrational modes are at 1699, 1650, 1444, 1180, 1088, and 920 cm^−1^ (see Table 2 ❶). These wavenumber bands are assigned to asymmetric stretching of C=N, deformation of NH, NH_2_ rocking, and C-COO^-^ stretching, respectively. With a smaller impact, GLN and LYS also contribute to the absorption of LYZ toward the silver surface (see Table 2 ❷ and ❺). Here, C=O stretching (1615 cm^−1^), NH and NH_3_^+^ deformation (1569, 1144 cm^−1^), and CCN stretching (954 cm^−1^) are the characteristic SERS peaks of GLN. In the case of LYS, due to the probable contact of its long side chain with the silver surface, CH_2_ scissoring (1460 cm^−1^), CH_2_ wagging (1273 cm^−1^), and CC/CN stretching (885 cm^−1^) are the most prominent peaks assigned in the SERS spectra. Moreover, in IL-8, LYS plays a minimal role in the interaction with the silver surface, together with GLU (see Table 2 ❸). This is characterized by the SERS signals at 1460, 1273, and 1242 cm^−1^, derived from CH_2_ scissoring and NH/CH_2_ wagging vibrational modes [40,41]. Instead, for the adsorption process of IL-8, the main driver during the adsorption is ASN (see Table 2 ❹) with its major bands at 1585, 1325, and 954 cm^−1^, assigned to COO^−^ asymmetric stretching, deformation CH_2_ and CCN stretching vibration, respectively [42,43].

## 4. Discussion

Saliva contains mostly water plus a variety of inorganic compounds, glycoproteins from mucus, epithelial cells, blood cells, and enzymes, notably, lysozyme. SERS spectra recorded on saliva samples are complex and show broad peaks resulting from many closely spaced peaks owned by multiple chemical species. It is composed mainly of contributions from vibrational modes originating from amide bands and aromatic and non-aromatic amino acid side chain residues. Thus, the most pronounced SERS signals are assigned to proteins. 

Here, we take into account one of the most studied salivary protein biomarkers, IL-8, and the LYZ bacteriolytical enzyme whose dysregulated expression (high concentration) has been used for the diagnosis of OSCC [31,32,33] and possible oral infections, respectively [35,36]. Based on MD simulations, we found that these proteins exhibit a high level of structural stability during interaction with and adsorption on the silver surface, resulting in no significant conformational rearrangements and therefore no loss of SERS signal information. Both IL-8 and LYZ have shown a diffuse banding mode, which allows the proteins to adapt flexibly to the surface, minimizing the impact on their tertiary structure. Despite general preservation of the tertiary structure in both cases, a small perturbation of the secondary structures is observed, suggesting a different behavior between alpha and beta folding. The interaction with the silver surface promotes the unfolding of the ordered alpha-helixes into random coils, which can form closer contact with the SERS substrate. This interaction is mainly driven by large side chains of amino acids (e.g., ARG and LYS) that selectively favor a parallel arrangement on the surface. This maximizes the number of contacts and thus stabilizes the complex. In particular, the driver of the adsorption resulted in ARG for LYZ and ASN and LYS for IL-8. Using the correlation between SERS and MD, we evaluated the discriminative SERS bands responsible for the differentiation between the two proteins LYZ and IL-8 to be the ARG signals for LYZ and in the ASN bands for IL-8. 

This study was performed as a proof-of-concept demonstration to show the feasibility of using the label-free SERS technique in combination with MD simulations to deeply understand the role of these biomarkers in the diagnosis of OSCC and oral infection with a high level of sensitivity and specificity. The presence of a dramatic increase in the concentration of LYZ and IL-8 due to OSCC or oral infections will lead to an augment in the relative intensity of the characteristic SERS peaks of the two identified amino acids, ARG and ASN, involved in the adsorption of the two proteins, LYZ and IL-8, on the silver surface.

From a technical point of view, the presented combined MD-SERS tool offers many advantages such as high sensitivity, low detection limits, low cost, simple design, and ease of sample collection, showing a tremendous promise for use in medical diagnostics. In our future work, in order to validate our approach, we plan to introduce a large patient cohort that has a statistically significant number of OSCC cancer and controls along with orally infected patients. With this cohort, we will be able to implement a distinctive and robust machine learning algorithm that can, with the help of the SERS spectra and MD calculations, classify the saliva biomarker as an OSCC cancer or oral infection one. Furthermore, numerous different salivary constituents are suggested as potential salivary biomarkers for oral cancer and oral infection, making the application of biomarkers for detection a challenging prospect. Most of these potential biomarkers are proteins and are present in low concentrations and can therefore be only detected with a highly sensitive approach. In our plans, we want to extend our approach to other biomarkers, largely studied in the literature, which are related to other oral cavity diseases.

## 5. Conclusions

In conclusion, we describe a proof-of-concept study in which the unique features of two techniques, i.e., SERS and MD, are successfully combined to improve salivary biomarker detection and shed light on their role in the perspective of clinical diagnosis of OSCC and oral infection. MD simulations reveal the underlying biomolecular mechanism of disease-related marker adsorption on silver surfaces that contribute to understanding the SERS response for LYZ and IL-8. Our findings suggest that this combined method might help to recognize disease features. Finally, with the use of the newest portable and handheld Raman devices, it would be significant to perform longitudinal studies using saliva periodically sampled from the same patient cohort to evaluate whether our findings correlate with disease stages.

## Figures and Tables

**Figure 1 sensors-23-08915-f001:**
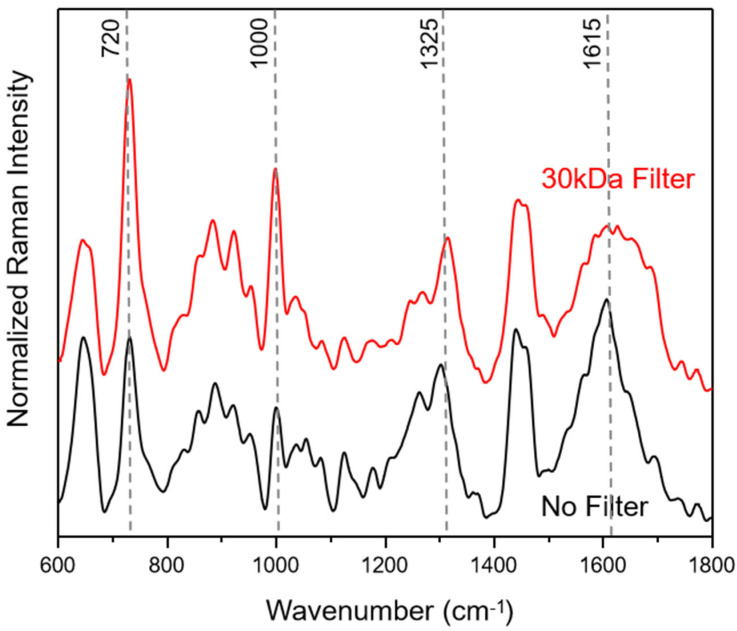
Average saliva SERS spectra before (black) and after filtering (red) with a 30 kDa cut-off. The median saliva SERS spectra are calculated from 250 individual spectra for each of the three healthy volunteers.

**Figure 2 sensors-23-08915-f002:**
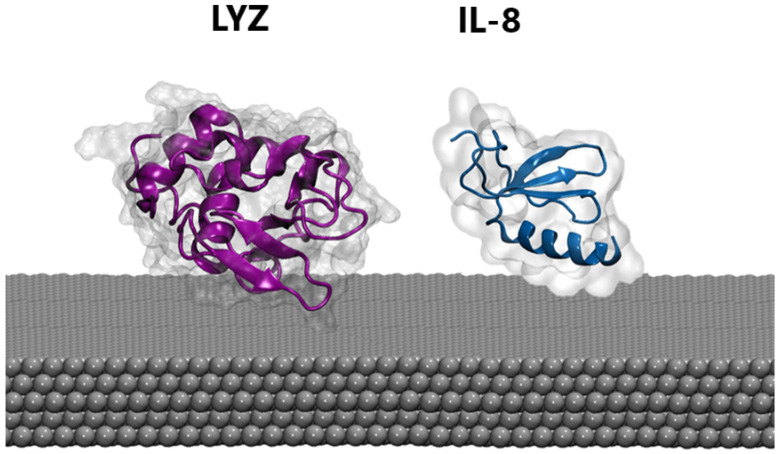
The most representative binding poses of LYZ (purple) and IL-8 (blue) protein on the silver surface.

**Figure 3 sensors-23-08915-f003:**
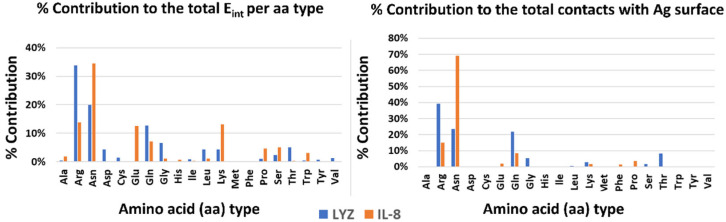
Energetics and dynamics contribution per amino acid type to the stabilization of the protein–silver complexes.

**Figure 4 sensors-23-08915-f004:**
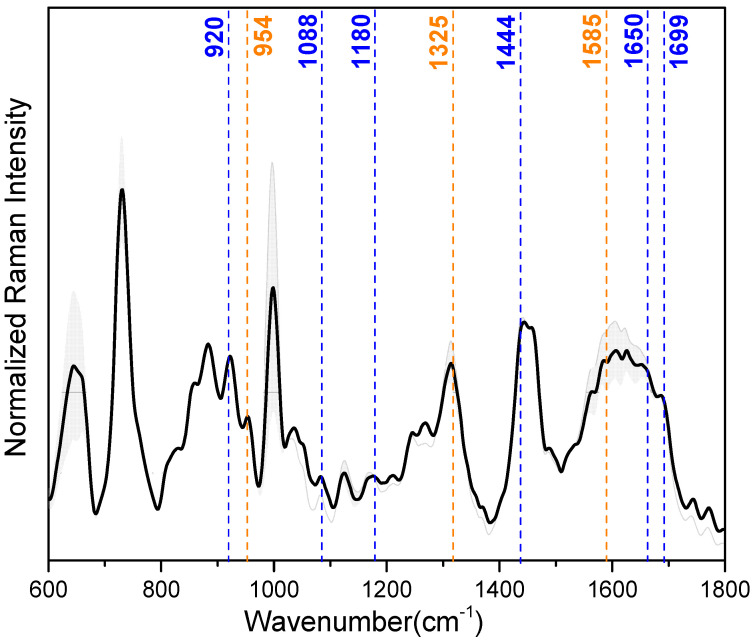
Median (solid lines) and standard deviation (SD) range (shaded areas) of the baseline-corrected and intensity-normalized SERS spectra of 30 kDa-filtered saliva samples. Dash lines highlight the specific SERS vibrational modes from the amino acids of LYZ (blue) and IL-8 (orange) (see also Table 2).

**Table 1 sensors-23-08915-t001:** Interaction energy calculated for the whole protein (E_int_) and divided by the number of protein residues (E_int_/resid) and secondary structure properties. The difference compared to the values calculated without the silver surface is given in parentheses.

	E_int_	E_int_/Resid	Secondary Structure Propensity		
	(kcal mol^−1^)	(kcal mol^−1^)	Alpha	Beta	Coil
LYZ	−15.6 (±1.2)	−0.12	39% (−7)	11% (0)	51% (+7)
IL−8	−8.6 (±1.6)	−0.13	24% (−4)	27% (−1)	49% (+5)

**Table 2 sensors-23-08915-t002:** Identified SERS peaks attributed to the most relevant amino acids (ARG, GLN, GLU, ASN, and LYS) involved in the absorption of LYZ and IL-8, based on data reported by Botta et al. [40], Barucci et al. [41] and Aliaga et al. [42,43] (±8 cm^−1^). The wavenumber in each row of the first column is correlated with a specific vibrational band in the last column assigned to one or more amino acids involved in the LYZ and IL-8 absorption on the silver surface.

Wavenumber (cm^−1^)	ARG	GLN	GLU	ASN	LYS	Tentative Band Assignment
1699	❶					ʋas C=N, ʋas COO^−^, δas NH_3_^+^
1650	❶	❷				Amide I
1615		❷				C=O str.
1585				❹	❺	COO^−^ asymmetric stretching
1569		❷				Def. NH
1460			❸		❺	CH_2_ sciss.
1444	❶					Def. NH vibr.
1325				❹		Def. CH_2_
1273			❸		❺	NH/CH_2_ wagging
1242			❸			CH_2_ wagging
1180	❶					NH_2_ rocking vibration
1144		❷				NH_3_^+^ deformation
1088	❶					NH deformation
954		❷		❹		CCN stretching
920	❶					C-COO^−^ stretching
885					❺	C-C/C-N stretching
817			❸			CC skeletal stretch.
728	❶	❷	❸	❹	❺	COO^−^ def.
654		❷	❸	❹		COO^−^ wag.
	LYZ	LYZ	IL-8	IL-8	IL-8/LYZ	

## Data Availability

Not applicable.

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
