# Peer review of "Label-Free SERS and MD Analysis of Biomarkers for Rapid Point-of-Care Sensors Detecting Head and Neck Cancer and Infections"

_sensors, 2023, doi:10.3390/s23218915_

Round 1
Reviewer 1 Report
See the attached file.

Author Response
In the following article, an innovate approach combining SERS technology and molecular dynamics simulations is reported as a powerful method for the noninvasive detection of salivary molecules recognized as biomarkers for head neck cancer and infections. To demonstrate the feasibility of the approach, the authors analyzed real salivary samples that were treated following a protocol, aiming at verifying the traceability of two types of proteins. The work paves the way for the development of sensitive non-invasive diagnostic devices characterized by high portability. The paper is well written, however, I have some comments I would like the authors to answer:
We thank reviewer #3 for the positive response and the comments are addressed below.
- I would like the authors to add information on how the two salivary markers are currently detected and, if they exist, what are the clinical standard.
In clinical research, both the biomarkers are mainly detected through ELISA test. In the last few years, there have been some spectroscopy test for detecting these markers in body fluids, with the purpose to reduce the cost and increase the sensitivity.
We have added a phrase (highlighted in red) and a citation in the introduction section as suggested from the reviewer:
“Currently detected predominantly by means of expensive enzymatic immunoassay (i.e ELISA test) [30], […].”
[30] O'Neill, C. D., et al. "Salivary concentrations of IL-8 and IL-1ra after HIIT and MICT in young, healthy adults: A randomized exercise study." Cytokine 157 (2022): 155965.
To the best of our knowledge, for these salivary markers, i.e. lysozyme and IL-8, no clinical standard values have been approved.
2) What are the range of concentrations of the two salivary biomarkers both in physiological and pathological conditions?
From the papers we have cited in the manuscript (references 31 until 37), the concentrations of these biomarkers in physiological status are in the range of 0.5-4 ng/mL for lysozyme and 1-3 ng/mL for IL-8.
In case of irregular physiological condition, the concentrations increase of several times. The range in these cases change by different factors (i.e. age, gender and kind of disease).
3) Are there any SERS approaches reported from the literature to detect these biomarkers?
We have cited related studies (references 14-22) where label-free SERS approach was applied for testing body-fluids with the purpose to discriminate between cancer patients and controls and to highlight specific vibrational bands for some biomarkers. Specifically, no papers are present in literature where these two salivary markers are detected by means of label-free SERS.
4) The MD simulations were implemented by taking into account all the boundary conditions of the experiment and among all, solvated conditions were mentioned, with the presence of water molecules. By the way, the SERS spectra were acquired after saliva samples drying on the Ag substrates. Did the simulations consider this drying step that could induce important conformation changes and protein unfolding?
In this context, the molecular dynamics technique is used as an integrative tool to the SERS measurement in order to detect the main conformational changes of the protein interacting with the silver surface. The final goal is to obtain a fast and reliable model that can possibly describe the main in vivo interactions and thus guide the spectral recognition. It is assumed that adsorption on the surface under wet conditions can explain the main determinants of the interaction. In order to make this simplification more realistic, the drying process is carried out slowly and at room temperature to minimize possible drastic conformational changes on the protein structure. Simulating the drying process of biological macromolecules on surfaces is an ambitious goal, but would require the implementation of computational techniques that are difficult to scale up and perform, and are beyond the scope of this work.
Same approach was used in other paper (reference 27) as cited (i.e. Wang, H., Xue, Z., Wu, Y., Gilmore, J., Wang, L., & Fabris, L. (2021). Rapid SERS quantification of trace fentanyl laced in recreational drugs with a portable Raman module. Analytical chemistry, 93(27), 9373-9382.).
Anyway, as pointed out by the referee, a limitation on this SERS/MD approach is present and a sentence (highlighted in red) have been added in the methods paragraph:
“This procedure allows the leaning of the silver nanopillars for a most intense SERS response and minimize the molecular rearrangements during the absorption of biomarkers on silver surface, in order to handle the limitation of MD simulations of being able to be performed only in waterly conditions.”
5) I would like to ask the authors if the Ag substrates were characterized in terms of resonant properties and eventual Raman background signal.
The silver SERS substrates used for our analysis are commercial available. They were purchased from Silmeco ApS (Denmark). These substrates have been largely used for many applications in literature where they report resonant properties and background signal. Below a list of some publications where they have provided these details:
Hakonen, Aron, et al. "Detecting forensic substances using commercially available SERS substrates and handheld Raman spectrometers." Talanta 189 (2018): 649-652.
Karawdeniya, Buddini Iroshika, et al. "General strategy to make an on-demand library of structurally and functionally diverse SERS substrates." ACS Applied Nano Materials 1.2 (2018): 960-968.
We have added the citations above within the Method section (references 38 and 39) when mentioning the SERS substrates purchased from Silmeco ApS (Denmark) for the first time.
- In Figure 4, the bands ascribable to the amino acids involved in the interaction with the silver were assigned. According to the literature, these amino acids are present in the structure of the two salivary targets. However, I was wondering if, after the 30 kDa cut-off filtration, no other small components are present in the saliva samples. How do you guarantee that only LYZ and IL-8 are present?
We want to point out that, even after 30 kDa cut-off filtration, the saliva contains other molecules in addition to LYZ and IL-8. In case of disease, infections or oral cancer, the concentration of these two salivary biomarkers notably increases, bringing them to occupy large part of the SERS hot-spots generated on the silver nanopillars surface. So, the SERS spectra collected from an infected or cancer patient, would be largely dominated by the vibrational contributions from one or both these markers. Moreover, as reported in Figs S2 and S3, both lysozyme and interleukin-8 show a notable affinity and stability in the absorption on silver nanoparticles.
7) Following the previous question, would it be useful to acquire spectra of simplified salivary models (e.g. LYZ diluted in water)? Or to perform the analysis on enriched samples?
We are going to plan analyses on enriched samples, in order to evaluate the interference of other salivary components in the SERS measurements, but as we have already pointed out in the previous question, LYZ and IL-8 show a significant affinity and stability in the absorption on silver nanoparticles surface.
8) In the discussion section, the authors stated that “From a technical point of view the presented combined MD-SERS tool offers many advantages such as high sensitivity, low detection limits, low cost, simple design and ease of sample collection, showing a tremendous promise for use in medical diagnostics” However, throughout the text no values are reported both on the range of concentrations of the salivary biomarkers to be detected and on the sensitivity of the proposed method. It would be interesting to have some numbers. In addition, the SERS spectra are acquired using confocal Raman microscope and objectives, not exactly low-cost components.
We want to stress that this proof-of-concept study has the purpose to demonstrate the feasibility of using the combination between label-free SERS technique and MD simulations to understand the role of these biomarkers in the diagnosis of cancer and oral infection with a high level of specificity. The analysis focuses on qualitative and not quantitative information from the SERS spectra, in fact we evaluate the differences in the relative intensity ratios of some Raman bands belonging to the biomarkers analyzed. Anyway, the information regarding the ranges of concentration of LYZ and IL-8, reported in the papers cited (reference 31 to 37), have been added in the introduction (highlighted in red).
Regarding the high cost of the Raman setup, we agree with the referee, but it is largely recognized that a Raman analysis results more cost-efficient, in terms of time and lab consumables, than the enzymatic immunoassays used in the clinics, i.e. ELISA test.
Moreover, the recent handheld Raman systems, like the one used in the paper of Wang et al. (reference 27), allow to reduce the costs per test. Also, these are easy to handle, being small and compact, and show a high-sensitivity. A sentence has been added in the conclusion (highlighted in red):
“Finally, through the use of the newest portable and handheld Raman devices, […].”
Reviewer 2 Report
The manuscript entitled "Label-free SERS and MD analysis of salivary biomarkers for rapid POC sensors detecting head neck cancer and infections" can be reconsidered and revised carefully for possible publication in the journal.
1. The aim of study is not obvious and should be rewritten.
2. The sample size has not been mentioned.
3. FE-SEM imaging of SERS should be provided before and after modification
4. Raman wavenumbers and bands should be evaluated?
5. in which aim the amino acids in table 2 were provided. The explanation is required to provide
6. The wave number didn't inform in table 2. the caption of table 2 should be revised.
7. The table of similar study will be provided and the advantages and disadvantages compared to this study will be provided.
8. The limitation of this study should be mentioned. The conclusion should be divided from discussion.
Minor editing of English language required
Author Response
The manuscript entitled "Label-free SERS and MD analysis of salivary biomarkers for rapid POC sensors detecting head neck cancer and infections" can be reconsidered and revised carefully for possible publication in the journal.
We thank the reviewer #1 for the valuable comments, which we have addressed below.
- The aim of study is not obvious and should be rewritten.
We aim to shed light on the interaction of lysozyme (LYZ) and interleukin-8 (IL-8) with metallic nanostructures by a combined SERS-MD approach, which was not reported before. LYZ and IL-8 play an important role as cancer markers in biofluids. We revised the last paragraph of the introduction and believe, that the aim of the study is clearly described.
- The sample size has not been mentioned.
In the Methods section (highlighted in red) we have already provided the sample size and the procedure pre- and post- saliva sampling in order to test pure saliva. We collected saliva samples from 3 individuals. In Fig. 1 and Fig. S1, we have also reported all the SERS spectra collected for each of the saliva providers.
- FE-SEM imaging of SERS should be provided before and after modification
The silver SERS substrates used for our analysis are commercially available. They were purchased from Silmeco ApS (Denmark). These have been largely utilized in a lot of research applications where they report the FE-SEM images before and after modification of the silver nanopillars. Below a list of some publications where they have provided the FE-SEM images:
Hakonen, Aron, et al. "Detecting forensic substances using commercially available SERS substrates and handheld Raman spectrometers." Talanta 189 (2018): 649-652.
Karawdeniya, Buddini Iroshika, et al. "General strategy to make an on-demand library of structurally and functionally diverse SERS substrates." ACS Applied Nano Materials 1.2 (2018): 960-968.
We have added the citations above within the Method section (references 38 and 39) when mentioning the SERS substrates purchased from Silmeco ApS (Denmark) for the first time.
- Raman wavenumbers and bands should be evaluated?
The Raman modes and their spectral positions of LYZ and IL-8 were compared with data from the literature in order to confirm the correct assignment of the marker modes. An assignment of the Raman modes is provided in Table 2.
- in which aim the amino acids in table 2 were provided. The explanation is required to provide
In Table 2, we reported Raman bands attributed to the most relevant amino acids involved in the absorption of Lysozyme and Interleukin-8 on the silver surface. Based on previous studies, we identified the most important vibrational modes for each of the amino acids selected (arginine, glutamic acid, asparagine, glutamine and lysine). These Raman peaks will be fundamental for recognize the presence of high concentration of Lysozyme and Interleukin-8 in saliva collected from patients with alterations in the physiological status, i.e. infections or cancer, in the worst case. We improved the caption of Table 2 as indicated in the manuscript.
- The wave number didn't inform in table 2. the caption of table 2 should be revised.
In Table 2, through the study of previous publications in literature, each wavenumber is correlated to a specific vibrational mode, assigned on the basis of the importance for any specific amino acids owned to these biomarkers taken into consideration, during the absorption on silver SERS nanoparticles. The caption of this table has been modified in order to make it more easily readable. The corrections are highlighted in red.
- The table of similar study will be provided and the advantages and disadvantages compared to this study will be provided.
To date, our study results are the first where SERS and MD are combined in order to recognize the patterns of two salivary constituents in real samples for point-of-care diagnostics. In literature there are only two papers in which the SERS/MD approach was applied but they are focused on distinguishing different kind of drugs in urine [Ding et al., Wang et al.]. We believe that a table as suggested would confuse the reader because in the other publications the identification of external substances introduced in human body-fluids were aimed and not metabolites produced by our own organism in case of human status alterations, as we aimed to do. Below the references above-mentioned are listed, both papers are already cited in the paper (references 27 and 28).
Ding, Z., Wang, C., Song, X., Li, N., Zheng, X., Wang, C., ... & Liu, H. (2023). Strong π-Metal Interaction Enables Liquid Interfacial Nanoarray–Molecule Co-assembly for Raman Sensing of Ultratrace Fentanyl Doped in Heroin, Ketamine, Morphine, and Real Urine. ACS Applied Materials & Interfaces, 15(9), 12570-12579.
Wang, H., Xue, Z., Wu, Y., Gilmore, J., Wang, L., & Fabris, L. (2021). Rapid SERS quantification of trace fentanyl laced in recreational drugs with a portable Raman module. Analytical chemistry, 93(27), 9373-9382.
- The limitation of this study should be mentioned. The conclusion should be divided from discussion.
A sentence in methods paragraph (highlighted in red), regarding the limitation of this study, has been written:
This procedure allows the leaning of the silver nanopillars for a most intense SERS response and minimize the molecular rearrangements during the absorption of biomarkers on silver surface, in order to handle the limitation of MD simulations of being able to be performed only in waterly conditions.
Moreover, a conclusionary paragraph has been added:
In conclusion, we describe a proof-of-concept study in which the unique features of two techniques, i.e. SERS and MD, are successfully combined to improve salivary biomarkers detection and shed light into their role in perspective of clinical diagnosis of OSCC and oral infection. MD simulations reveal the underlying biomolecular mechanism of disease-related markers adsorption on silver surfaces that contributes to understand the SERS response for LYZ and IL-8. Our findings suggest that this combined method might help to recognize disease features. Finally, through the use of the newest portable and handheld Raman devices, it would be significant to perform longitudinal studies using saliva periodically sampled from the same patient cohort to evaluate whether our findings correlate with disease stages.
Reviewer 3 Report
The present manuscript entitled "Label-free SERS and MD analysis of salivary biomarkers for rapid POC sensors detecting head neck cancer and infections" by Edoardo Farnesi, Silvia Rinaldi, Chen Liu, Jonas Ballmaier, Orlando Guntinas-Lichius, Michael Schmitt, Dana Cialla-May, and Juergen Popp (sensors-2616354) is written correctly and has a good structure; moreover, it has all the necessary parts. The article is interesting from a spectroscopy and biosensor point of view; therefore, it should interest the reader. I proposed a few improvements. The paper meets Sensors' requirements, and I recommend the article for publication in Sensors following the common editing stage. My current decision is a minor revision. More specific comments and observations are presented below.
1. Abstract. Information about the specific biomarkers that were focused on in the studies may be added.
2. Keywords. The abbreviation MD can be expanded - MD itself is not very informative. Keywords can be arranged alphabetically.
3. You can add more detailed information about healthy individuals, e.g., gender, age.
4. Results. How was the average SERS spectra calculated? For three samples? Or some other way?
5. Description of Figure 1. “A)” should be removed.
6. Figures 3, S2, and S3. It would be good to add the axis name and unit. Figure S1 - the most important bands can be assigned numerically as in Fig. 1.
7. Does the developed method have disadvantages?
8. Are studies on the determination of IL-8 and LYZ planned? What about interference analysis?
9. Conclusions should be distinguished as a separate section.
10. References should be adapted to the requirements of the journal. Currently, for example, the journals' names are sometimes written with full names and sometimes with abbreviations.
I hope that the comments presented will help improve the article.
Author Response
The present manuscript entitled "Label-free SERS and MD analysis of salivary biomarkers for rapid POC sensors detecting head neck cancer and infections" by Edoardo Farnesi, Silvia Rinaldi, Chen Liu, Jonas Ballmaier, Orlando Guntinas-Lichius, Michael Schmitt, Dana Cialla-May, and Juergen Popp (sensors-2616354) is written correctly and has a good structure; moreover, it has all the necessary parts. The article is interesting from a spectroscopy and biosensor point of view; therefore, it should interest the reader. I proposed a few improvements. The paper meets Sensors' requirements, and I recommend the article for publication in Sensors following the common editing stage. My current decision is a minor revision. More specific comments and observations are presented below.
We thank reviewer #2 for the positive response. We have addressed the comments below.
- Abstract. Information about the specific biomarkers that were focused on in the studies may be added.
We have specified the biomarkers in the abstract.
- Keywords. The abbreviation MD can be expanded - MD itself is not very informative. Keywords can be arranged alphabetically.
We have rearranged the keyword list properly.
- You can add more detailed information about healthy individuals, e.g., gender, age.
We have added the following table (Table S1) to the supplementary materials.
|
ID |
GENDER |
YOB |
STATUS |
|
#1 |
Male |
1972 |
Healthy |
|
#2 |
Male |
1969 |
Healthy |
|
#13 |
Female |
1962 |
Healthy |
- Results. How was the average SERS spectra calculated? For three samples? Or some other way?
In the Methods section we have provided the sample size and the procedure before and after saliva sampling in order to test pure saliva (highlighted in red). In Fig. 1 and Fig. S1, we have also reported all the SERS spectra collected for each of the saliva providers (250 spectra).
- Description of Figure 1. “A)” should be removed.
We have corrected the typo.
- Figures 3, S2, and S3. It would be good to add the axis name and unit. Figure S1 - the most important bands can be assigned numerically as in Fig. 1.
We have properly modified Figs 3, S1, S2 and S3 according to the suggestion.
- Does the developed method have disadvantages?
Our work is a proof-of-concept testing SERS/MD approach for detection of small salivary biomarkers. Nowadays, we can point out that simulating larger bio macromolecules with a molecular weight of over 50 kDa (i.e. human serum albumin in saliva or serum) would require a considerable amount of time, both in terms of simulation and data analysis. However, in the last few years, the most important biomarkers identified, especially in cancer diagnostics, tend to be relative small molecules, with molecular weights less than 30 kDa (i.e. cytokines, chemokines, etc.), facilitating the use of our combined method. Thus, we would like to stress that the larger the molecule is, the longer the simulation needs, which hinders their routine application. But since found marker molecules having a molecular weight lower than 30 kDa, this disadvantage does not come into play.
- Are studies on the determination of IL-8 and LYZ planned? What about interference analysis?
The determination of IL-8 and LYZ is our next future step, together with the analysis and comparison of samples from both patients and healthy providers. We would identify these two molecules but also other possible biomarkers, which can help the physicians in the diagnosis of disease. Obviously, an interference analysis will be taken into account.
- Conclusions should be distinguished as a separate section.
We have added a conclusionary paragraph as suggested.
- References should be adapted to the requirements of the journal. Currently, for example, the journals' names are sometimes written with full names and sometimes with abbreviations.
We have corrected the typos.
Round 2
Reviewer 2 Report
The most concerns have been addressed.